# The Role of Late Presenters in HIV-1 Transmission Clusters in Europe

**DOI:** 10.3390/v15122418

**Published:** 2023-12-13

**Authors:** Mafalda N. S. Miranda, Victor Pimentel, Perpétua Gomes, Maria do Rosário O. Martins, Sofia G. Seabra, Rolf Kaiser, Michael Böhm, Carole Seguin-Devaux, Roger Paredes, Marina Bobkova, Maurizio Zazzi, Francesca Incardona, Marta Pingarilho, Ana B. Abecasis

**Affiliations:** 1Global Health and Tropical Medicine (GHTM), Associate Laboratory in Translation and Innovation towards Global Health (LA-REAL), Institute of Hygiene and Tropical Medicine, New University of Lisbon (IHMT/UNL), 1349-008 Lisbon, Portugal; victor.pimentel@ihmt.unl.pt (V.P.); mrfom@ihmt.unl.pt (M.d.R.O.M.); sgseabra@ihmt.unl.pt (S.G.S.); martapingarilho@ihmt.unl.pt (M.P.); ana.abecasis@ihmt.unl.pt (A.B.A.); 2Laboratório de Biologia Molecular (LMCBM, SPC, CHLO-HEM), 1349-019 Lisbon, Portugal; pcrsilva@chlo.min-saude.pt; 3Centro de Investigação Interdisciplinar Egas Moniz (CiiEM), Instituto Universitário Egas Moniz, 2829-511 Costa da Caparica, Portugal; 4Institute of Virology, University Hospital of Cologne, University of Cologne, 50923 Cologne, Germany; rolf.kaiser@uk-koeln.de (R.K.); michael.boehm@uk-koeln.de (M.B.); 5DZIF, Deutsches Zentrum für Infektionsforschung, German Center for Infection Research, Partner Site Bonn-Cologne, 50923 Cologne, Germany; 6Laboratory of Retrovirology, Department of Infection and Immunity, Luxembourg Institute of Health, L-4354 Esch-sur-Alzette, Luxembourg; carole.devaux@lih.lu; 7Infectious Diseases Department, IrsiCaixa AIDS Research Institute, Hospital University Hospital Germans Trias i Pujol, 08916 Badalona, Spain; rparedes@irsicaixa.es; 8Gamaleya National Research Center of Epidemiology and Microbiology, 123098 Moscow, Russia; mrbobkova@mail.ru; 9Department of Medical Biotechnologies, University of Siena, 53100 Siena, Italy; maurizio.zazzi@unisi.it; 10IPRO—InformaPRO S.r.l., 00152 Rome, Italy; f.incardona@informa.pro; 11EuResist Network, 00152 Rome, Italy

**Keywords:** HIV-1 infection, late presenters, non-late presenters, transmission clusters

## Abstract

Background: Investigating the role of late presenters (LPs) in HIV-1 transmission is important, as they can contribute to the onward spread of HIV-1 virus before diagnosis, when they are not aware of their HIV status. Objective: To characterize individuals living with HIV-1 followed up in Europe infected with subtypes A, B, and G and to compare transmission clusters (TC) in LP vs. non-late presenter (NLP) populations. Methods: Information from a convenience sample of 2679 individuals living with HIV-1 was collected from the EuResist Integrated Database between 2008 and 2019. Maximum likelihood (ML) phylogenies were constructed using FastTree. Transmission clusters were identified using Cluster Picker. Statistical analyses were performed using R. Results: 2437 (91.0%) sequences were from subtype B, 168 (6.3%) from subtype A, and 74 (2.8%) from subtype G. The median age was 39 y/o (IQR: 31.0–47.0) and 85.2% of individuals were males. The main transmission route was via homosexual (MSM) contact (60.1%) and 85.0% originated from Western Europe. In total, 54.7% of individuals were classified as LPs and 41.7% of individuals were inside TCs. In subtype A, individuals in TCs were more frequently males and natives with a recent infection. For subtype B, individuals in TCs were more frequently individuals with MSM transmission route and with a recent infection. For subtype G, individuals in TCs were those with a recent infection. When analyzing cluster size, we found that LPs more frequently belonged to small clusters (<8 individuals), particularly dual clusters (2 individuals). Conclusion: LP individuals are more present either outside or in small clusters, indicating a limited role of late presentation to HIV-1 transmission.

## 1. Introduction

At the end of 2020, there were 37.7 million people living with HIV [1]. In HIV epidemics, certain risk groups contribute to the spread of HIV disproportionately more than others. This can be due to determinants of the host such as specific demographic, clinical, or behavioral factors or to viral determinants, such as the viral strain infecting the host [2,3]. On the one hand, the literature suggests that an undiagnosed recent infection could be associated with the transmission and spread of HIV-1 [4]. Furthermore, higher values of viral load in an individual living with HIV can also increase the likelihood of transmission, especially in individuals in the acute phase of infection [5]. On the other hand, late presentation to diagnosis has increased over the years and in Europe. Late presenters (LPs) account for around 50% of new HIV diagnoses [6]. The consensus definition of late presenters (LPs) says that these are individuals living with HIV-1 diagnosed with a baseline CD4 count lower than 350 cells/mm^3^ or with an AIDS-defining event, regardless of CD4 cell count [7]. Late presentation is associated with high morbidity and mortality, at an individual level, and increased health costs [8]. Furthermore, LPs can also contribute to the onward spread of HIV-1 virus at the population level, as these individuals are not aware of their HIV status and could also spread the virus without knowing their infection status [9]. 

The use of powerful tools such as phylogenetic trees and transmission clusters (TCs) is essential to understand the dynamics of viral transmission and to identify groups of individuals connected to each other [2,10].

In this study, we aim to describe the clinical and socio-demographic characteristics of individuals living with HIV-1 followed in Europe according to subtype and to understand the determinants associated with clustering on each of the most prevalent subtypes. Specifically, we performed an analysis of transmission clusters for HIV-1 subtypes, B, A, and G, and compared the patterns of transmission clusters in late presenter (LP) vs. non-late presenter (NLP) populations.

## 2. Materials and Methods

### 2.1. Study Group

We used a convenience sampling approach. 2679 individuals living with HIV- from the EuResist Integrated Database (EIDB) between 2008 and 2019 were included in this study. The EuResist Integrated Database (EIDB) is one of the largest existing databases which integrates clinical, socio-demographic, and viral genotypic information from individuals living with HIV-1. It collects longitudinal, periodically updated data mainly from Italian (ARCA database), German (AREVIR database) Spanish (CoRIS and IRSICAIXA), and Swedish, Belgian, Portuguese, and Luxembourgian databases [11].

In this study, information from the ARCA, AREVIR, Luxembourgian, IRSICAIXA, Portuguese, and Russian databases was used.

### 2.2. Drug Resistance Analysis and Subtyping

The HIV-1 *pol* sequences were derived from routine clinical genotypic resistance tests (Sanger method). The size of the reverse transcriptase (RT) and protease (PR) fragments used for this analysis was between 500 and 1000 nucleotides. Only the first HIV genomic sequence per individual was analyzed. Only ART-naïve individuals—classified as those who had a sample collection date for the first drug resistance test before the date of start of first therapy—were included. 

Transmitted Drug Resistance (TDR) was defined as the presence of one or more surveillance drug resistance mutations in a genomic sequence, according to the WHO 2009 surveillance list [12]. The sequences were submitted to the Calibrated Population Resistance tool version 8.0.

HIV-1 subtyping was performed using the consensus of the results obtained based on three different subtyping tools: REGA HIV Subtyping Tool version 3.46 (https://www.genomedetective.com/app/typingtool/hiv (accessed on 16 December 2021)) [13]; COMET: adaptive context-based modeling for HIV-1 (https://comet.lih.lu (accessed on 16 December 2021)) [14]; and SCUEAL (http://classic.datamonkey.org/dataupload_scueal.php (accessed on 16 December 2021)) [15].

### 2.3. Transmission Cluster (TC) Identification

For the analysis of transmission clusters and construction of phylogenetic trees, the database was divided in three separate datasets, corresponding to evolutionarily independent epidemics of subtypes A, B, and G. The control sequences were retrieved from the Los Alamos database, including all HIV-1 *pol* sequences from subtypes A, B, and G from Europe, South America, and Africa (http://www.hiv.lanl.gov (accessed on 8 November 2023)) and 38,531 other sequences from the EuResist database [16]. Three subtype B and C reference sequences retrieved from the Los Alamos database were used as an outgroup. For each subtype, the sequences were aligned against the control sequences dataset using VIRULIGN [17]. The HIV-1 K03455.1 (HXB2) *pol* nucleotide sequence (nt) was used as reference for the codon-corrected alignment. The dataset was then manually edited to exclude sequences with low quality, duplicates, and clones using the MEGA 7 software [18]. The final datasets of subtypes A, B, and G consisted of 10,122, 62,543, and 5547 sequences, respectively, with a length of 948 bp. Maximum likelihood (ML) phylogenies were constructed using FastTree [19] with the generalized time-reversible evolutionary model. Statistical support for clades was assessed using the Shimodaira–Hasegawa-like test (SH-test). Putative transmission clusters were identified using ClusterPicker v1.332 [20], using a definition of the threshold that included a genetic distance of 0.030 and a branch support ≥ 0.90 according to the approximate likelihood ratio test (aLRT). For analyses of cluster size, we defined clusters with 8 individuals or more as large clusters and cluster with less than 8 individuals as small clusters. The configuration of the phylogenetic trees was visualized using the iTOL software.

### 2.4. Study Variables

New variables were created according to: Migration status—Based on country of origin and country of follow-up (if country of origin and country of follow-up are the same, then the patient was classified as native; otherwise, as a migrant);Age at resistance test—Based on the difference between year of birth and date of the first drug resistance test;Region of origin—Based on country of origin;Recentness of infection—Based on ambiguity rate of genomic sequences. We defined chronic infection as the ambiguity rate being higher than 0.45%; otherwise, recent infection was defined, as previously described [21];LP vs. NLP at HIV diagnosis—Based on CD4 count, LPs were defined as individuals with a baseline CD4 count ≤ 350 cells/mm^3^ and NLPs were defined as individuals with a baseline CD4 count > 350 cells/mm^3^ [7].

### 2.5. Statistical Analysis

The proportion and median (interquartile range, IQR) were calculated for every categorical and continuous variable, respectively. Subtype variables were compared with the categorical variables using a Chi-square test, and continuous variables with a Mann–Whitney U test. Logistic regression was used to analyze the association between the clustering status for each subtype and demographic and clinical factors. First, we presented logistic regression with unadjusted odds ratios (uORs) and confidence intervals at 95% (95%CI); then, we included only the variables with a *p*-value < 0.05 in the final model. The final model was adjusted for sex: this variable was forced into the model regardless of its significance. Data were analyzed using RStudio (Version 1.2.5033).

## 3. Results

### 3.1. Characteristics of European Population of Individuals Living with HIV-1 

Among the 2679 individuals living with HIV-1 included in the analysis, 2437 (91.0%) were from subtype B, 168 (6.3%) from subtype A, and 74 (2.8%) were from subtype G. The median age at resistance test was 39.0 (31.0–47.0) years old and 85.2% of individuals were males. The main transmission route was via homosexual (men who have sex with men—MSM) contact (60.1%). For subtype B, the MSM route (64.2%) was also the most prevalent whereas the heterosexual route was the predominant route for subtypes A and G (50.8% and 69.2%, respectively). 85.0% of individuals originated from Western Europe and, according to subtypes, Eastern Europe was the most prevalent region of origin for subtypes B and A, whereas for G, the African region was the most prevalent. Most individuals included in this study were native (81.6%). However, while in subtype B, natives were more prevalent (95.4%), in subtypes A and G, migrants were more prevalent (19.7% and 12.8%). Based on the ambiguity rate of the first genomic sequence, most individuals were classified as presenting with a chronic infection (56.8%). The CD4 count at diagnosis and viral load at diagnosis (log10) presented a median of 320 cells/mm^3^ (IQR 134–506.5) and log10 4.7 copies/mL (IQR 4.3–5.4), respectively. Furthermore, 41.7% of the individuals in this population of patients clustered within transmission clusters and 54.7% were classified as LPs (CD4 < 350 cells/mm^3^). Most individuals from subtype B (95.6%) were within clusters, in contrast to subtypes A (3.5%) and G (1.5%), where a much lower percentage of patients was within clusters (Appendix A). 

### 3.2. Dynamics of Subtype A HIV-1 Epidemic in Europe

Based on the sequences from our database and the control sequences retrieved, we could observe that the majority of the subtype A population had its origin in Africa, and the major route of transmission was heterosexual. There were some individuals with IDU transmission. The phylogenetic analyses indicated that most EuResist individuals clustered into two different parts of the tree, indicated with arrows A and B in Figure 1, suggesting two parallel epidemics of subtype A in Europe. The first cluster included individuals originating from Africa and Eastern Europe via heterosexual and IDU transmission routes (cluster A) and the other cluster included individuals originating from Western Europe via MSM transmission (cluster B). LP individuals were mostly concentrated in cluster A, where the majority of individuals were also migrants (Figure 1).

### 3.3. Dynamics of Subtype B HIV-1 Epidemic in Europe

Based on the sequences from our database and the control sequences retrieved, we could observe that most subtype B individuals originated from Western Europe were natives and MSM. Individuals with IDU transmission originating from Western Europe dominated one cluster of the tree (indicated with an arrow). LP and NLP individuals were distributed evenly in the tree. Based on the configuration of the phylogenetic tree and, apart from the cluster dominated by IDU transmission, there are no major patterns of compartmentalization in the subtype B epidemic in Europe (Figure 2).

### 3.4. Dynamics of Subtype G HIV-1 Epidemic in Europe

Based on the sequences from our database and the control sequences retrieved, we could observe that two major regions of origin—Western Europe and Africa—contribute to the subtype G epidemic of HIV-1 in Europe. These are largely divided into two major clusters indicated with arrows A and B in Figure 3. Most individuals are heterosexual and late presentation dominates. The tree configuration indicates a lack of compartmentalization of the subtype G epidemic in Europe and suggests frequent importations of subtype G.

### 3.5. Determinants Associated with Transmission Clusters of HIV-1 in Europe for Different Subtypes

In the first unadjusted logistic regression model for subtype A, the variables associated with patients being in clusters from subtype A were male gender (*p* = 0.001), the MSM (*p* = 0.005) route of transmission, region of origin being Africa (*p* = 0.032), being native (*p* = 0.007), having a recent infection (*p* = 0.002), and being a NLP (*p* = 0.018) (Appendix A). In the final adjusted logistic regression model for subtype A, male individuals were more likely to be in clusters when compared to females (OR: 4.72, *p* = 0.002). Individuals with a recent infection were more likely to be in clusters when compared to individuals with a chronic infection (OR: 3.31, *p* = 0.008) and natives were also more likely to be in clusters when compared to migrants (OR: 2.43, *p* = 0.044) (Table 1).

In the subtype B unadjusted logistic regression model, the variables associated with the chance of patients being within clusters were male gender (*p* = 0.037), the MSM route of transmission (*p* < 0.001), having a recent infection (*p* < 0.001), and being a NLP (*p* < 0.001) (Appendix A). In the final adjusted logistic regression model, the MSM transmission route compared to the heterosexual route (OR: 1.50, *p* = 0.001) and recent infection compared to individuals with a chronic infection (OR: 2.29, *p* < 0.001) were associated with a higher chance of being in clusters (Table 1). 

In the subtype G unadjusted logistic regression model, the variables associated with patients being within clusters were region of origin being Africa (*p* = 0.018), being native (*p* = 0.033), and having a recent infection (*p* = 0.011) (Appendix A). In the final adjusted logistic regression model, recent infection was associated with clustering when compared to chronic infection (OR: 4.38, *p* = 0.012) (Table 1).

### 3.6. Size of Transmission Clusters

Of the 2679 individuals living with HIV, 1116 were within clusters (41.7%). The minimum cluster size was 2 and the maximum cluster size was 18.

The proportion of late presenters and non-late presenters in clusters was analyzed and both LPs and NLPs were more frequently in small clusters (93.7%; 92.7%, respectively). Also, LPs within the small clusters were more frequently in dual clusters (62.7%) (two individuals per cluster) (Figure 4A). 

According to subtypes, subtype A had a higher proportion of individuals in dual clusters (53.8%), subtype B had a higher proportion of individuals in clusters of 9 or more (6.1%), and subtype G had a higher proportion of individuals in clusters with size between 3–5 individuals and 6–8 individuals (36.4% and 8.1%, respectively) (Figure 4B).

### 3.7. Transmission Clusters in LP vs. NLP

Here, we compared the characteristics of LPs vs. NLPs within clusters. There were specific clinical and socio-demographic characteristics of LPs in clusters and NLPs in clusters. LPs were mainly out of clusters in all subtypes. 

In subtype A, the variables associated with being a LP vs. NLP within clusters were age at resistance test, recentness of infection, and viral load at diagnosis. The NLP individuals in clusters were aged between 19–31 y/o (*p* = 0.014), with a recent infection (*p* = 0.011), and with lower viral load values. 

In subtype B, LPs vs. NLPs within clusters were associated with age at resistance testing (*p* < 0.001), recentness of infection (*p* < 0.001), viral load at diagnosis (*p* < 0.001), and migration status (*p* = 0.025). LPs within clusters were females (*p* = 0.026), individuals older than 31 y/o (*p* = 0.026; *p* < 0.001), originated from South America (*p* = 0.011), with a viral load at diagnosis higher than 5.1 copies/mL (*p* < 0.001), presenting a chronic infection (*p* < 0.001), and with migrant status (*p* = 0.001). Meanwhile, the NLPs in the clusters were younger than 30 y/o (*p* < 0.001) and had a recent infection (*p* < 0.001) and a viral load at diagnosis lower than 4.0 copies/mL (*p* < 0.001). 

In subtype G, LPs vs. NLP distribution in the clusters was associated with the sex variable (*p* = 0.046). LPs in the clusters were mainly female (*p* = 0.046) and originated from Africa (100%) (Table 2).

## 4. Discussion

In the current work, we aimed to study and characterize HIV-1 transmission clusters in Europe with the specific objective of understanding the role of LPs in transmission of infection. Specifically, we aimed to understand HIV-1 transmission clusters and the determinants associated with transmission in clusters, taking into account the independent pandemics of the most prevalent subtypes in our population, A, B, and G, and understanding the clustering patterns of LPs and NLPs in each subtype. For that purpose, we used an integrated dataset of genomic sequences from the EuResist database and genomic sequences from public databases. HIV-1 transmission patterns were studied via the reconstruction of transmission clusters integrated with comprehensive clinical and socio-demographic data. This approach is highly useful for public health purposes, to fine-grain transmission patterns at a higher resolution compared to classical epidemiology. 

One important finding of our study showed incongruence between the definition of LPs based on CD4 counts and on the recentness of infection (based on sequence ambiguity rates). While both definitions should reflect recentness of infection, we found some cases of incongruence between the two classifications. These findings should be further considered in later investigations.

In our population, the majority of individuals were from subtype B and male with the MSM route of transmission and their region of origin being Western Europe. These results are in accordance with a previous study conducted in Europe to analyze the distribution of subtypes [3]. On the other hand, there were more individuals outside TCs (58.3%) compared to those inside TCs (43.2%) [22]. 

For subtype B, one of the factors identified as associated with being inside a cluster was the MSM transmission route [23]. For subtype A, we also found the MSM transmission route as a factor associated with being inside clusters in the univariate analysis, although it was not present in the final logistic model. The fact that the MSM route of transmission is associated with clustering in non-B subtypes is in accordance with several studies that report an increase in non-B subtypes transmission among MSMs [23,24,25]. Nevertheless, we expected an association of the IDU route of transmission and subtype A, given the characteristics of the epidemic in Eastern Europe [26]. 

Regarding the potential association between transmission clusters and late presentation, we found that both LPs and NLPs were mainly outside clusters. For those that were inside clusters, we compared the characteristics of the populations of LPs and NLPs, and these patterns were consistent between subtypes B and G. Concerning sex, there were more LP females inside clusters. In subtypes A and B, there were differences concerning age, recentness of infection, and viral load. In subtypes A and B, younger individuals were associated with being NLPs inside clusters, and this is in accordance with other studies associating younger age with clustering [22,27]. Furthermore, in subtype B, the LPs inside clusters were older (more than 31yo), with a higher and growing proportion of LPs in clusters as age increased. This extends previous findings where late presentation is associated with older age [6,8] by showing that LPs with an older age contribute more actively to HIV-1 transmission clusters. 

One interesting finding was that the LPs located inside clusters had higher viral loads than NLPs inside clusters. While no such analyses have been carried out concerning clustering, these results are congruent with other studies, which report higher viral loads in LPs when compared to NLPs [22,28]. 

Regarding migration status, for subtype B, a higher proportion of migrants with late presentation were found inside clusters. For the same subtype, we found the same pattern for individuals originating from South America. Subtype B had a higher proportion of migrants from Brazil, which has a concentrated HIV epidemic among MSMs [28,29]. This could indicate a shift in cluster composition for migrants from South America in recent years. While these are considered LPs in our analyses, we cannot discard the possibility of an unknown previous positive diagnosis in the country of origin later being considered a new diagnosis in the host country. For subtype G, most of the LPs inside clusters were from Africa, which was expected given the known epidemic distribution of this subtype.

Interestingly, for subtype A, migration status was also associated with clustering patterns. In this case, migrants infected with subtype A were less likely to be inside transmission clusters. In some European countries, HIV-1 subtype A infections have been associated with migration [30,31]. However, these results indicate that these migrants are not necessarily contributing to the dispersion of the pandemic. 

Finally, we found that LPs were more frequently present in small clusters or outside clusters, which suggests the limited role of this population on HIV-1 transmission. However, the higher viral loads observed in LPs located inside clusters can indicate the higher transmissibility of infection for those individuals inside TCs. 

There is still scarce to no information regarding transmission clusters and late presentation. We studied here the association of transmission clusters according to subtypes in LPs and NLPs, and our results showed that the patterns of LPs vs. NLPs in TCs presented some similar characteristics between the studied subtypes.

## 5. Conclusions

Our study highlights the patterns of transmission clusters in LP vs. NLP populations selected in the European dataset of EuResist. While late presentation was found to have a limited role on HIV-1 transmission, those that were inside clusters could have increased HIV transmissibility due to their higher viral loads. On the other hand, given our findings concerning the incongruence between recentness of infection (based on sequence ambiguity rate) and LP definition, further research should be conducted to reduce LP classification bias.

### Limitations

Despite the large size and coverage of the EuResist database, a major limitation is that it is based on a convenience sampling approach. As such, we must acknowledge the potential bias in our results due to the sample used.

As our work focused mainly on the LP vs. NLP populations, we also faced problems in using the current consensus definition of LPs. For that reason, we considered the definition of late presentation to be a limitation of the study since it is known that when an individual is acutely infected, the CD4 cell count drops before becoming higher again, which means that if they are diagnosed in that time period, i.e., at the acute stage, the patient can be misdiagnosed as late presenter. Furthermore, the accuracy of reconstructed transmission clusters is obviously affected by sampling issues and by the methods used for the reconstruction of phylogenetic trees. In our study, the use of the EuResist database may also have a limitation concerning the considerable amount of missing information for some variables. We managed to overcome this issue by considering for the different variables the absolute number of individuals as being those who had that information available. 

## Figures and Tables

**Figure 1 viruses-15-02418-f001:**
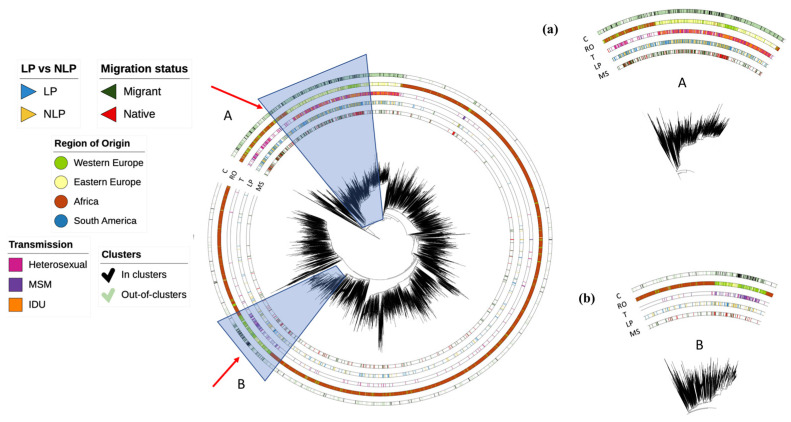
Phylogenetic tree for subtype A. This image shows a visual phylogenetic tree of the subtype A population. (**a**,**b**) are zoom images of the shadow zones A and B, respectively. Shadows A and B show the two portions/monophyletic clusters of the phylogenetic tree where the large majority of patients in our population clustered. Shadow A shows one major cluster of individuals originating from Africa and Eastern Europe with heterosexual and IDU transmission. Shadow B shows a cluster of individuals originating from Western Europe and with MSM transmission. C—clusters; RO—region of origin; T—transmission; LP—late presenters vs. non-late presenters; MS—migration status.

**Figure 2 viruses-15-02418-f002:**
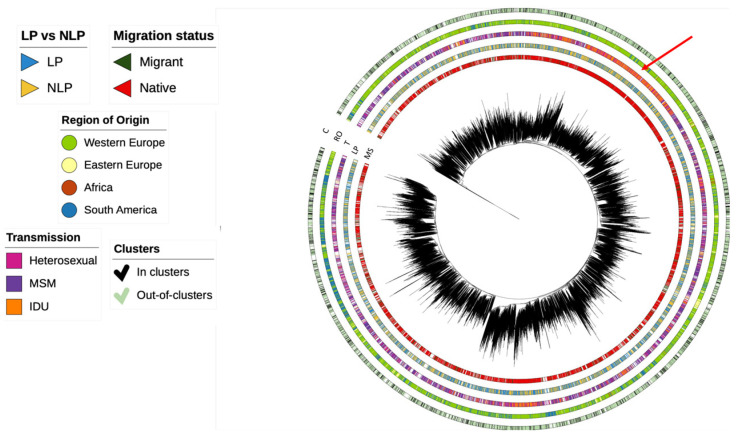
Phylogenetic tree for Subtype B. Visualization of the phylogenetic tree of subtype B. The region highlighted with an arrow indicates a monophyletic cluster of individuals with IDU transmission and originating from Western Europe. C—clusters; RO—region of origin; T—transmission; LP—late presenters vs. non-late presenters; MS—migration status.

**Figure 3 viruses-15-02418-f003:**
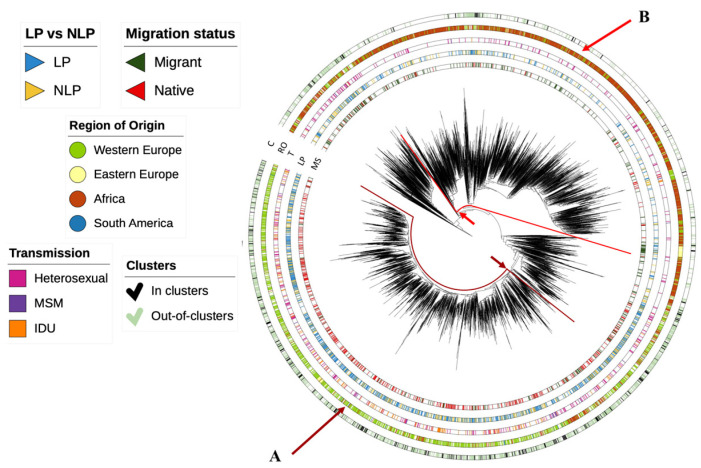
Phylogenetic tree for subtype G. This image shows a visual phylogenetic tree of the subtype G population. The regions highlighted with arrows shows monophyletic clusters dominated by individuals originating from Western Europe (A) and Africa (B). C—clusters; RO—region of origin; T—transmission; LP—late presenters vs. non-late presenters; MS—migration status.

**Figure 4 viruses-15-02418-f004:**
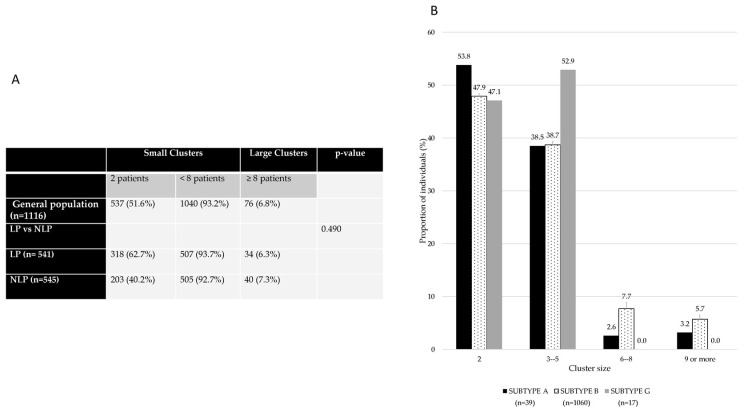
Characterization of cluster size according to late presentation (**A**) and cluster size according to subtype of infection (**B**). A cluster size <8 includes dual clusters (2 patients).

**Table 1 viruses-15-02418-t001:** Determinants associated with belonging to a transmission cluster according to subtypes A, B, and G.

		Subtype A		Subtype B		Subtype G	
In Clusters/Out of Clusters		Final Model (Stepwise)		Final Model (Stepwise)		Final Model (Stepwise)	
		aOR (95%CI)	*p*-Value	aOR (95%CI)	*p*-Value	aOR (95%CI)	*p*-Value
Sex	Female	Ref	Ref	Ref	Ref	Ref	Ref
	Male	4.72 (1.78–12.51)	0.002	1.03 (0.72–1.47)	0.877	1.74 (0.55–5.47)	0.345
Transmission Route	Heterosexual			Ref	Ref		
	MSM			1.50 (1.17–1.92)	0.001		
	IDU			1.50 (0.92–2.46)	0.106		
	Other			1.54 (0.60–3.94)	0.365		
Migration Status	Migrant	Ref	Ref				
	Native	2.43 (1.02–5.77)	0.044				
Recentness of Infection	Chronic	Ref	Ref	Ref	Ref	Ref	Ref
	Recent	3.31 (1.37–8.01)	0.008	2.29 (1.89–2.78)	<0.001	4.38 (1.38–13.65)	0.012
LP vs. NLP	LP	Ref	Ref				
	NLP	1.99 (0.84–4.70)	0.117				

**Table 2 viruses-15-02418-t002:** Characteristics of LPs and NLPs in clusters according to subtypes.

	In Clusters			
	LP	NLP	*p*-Value	*p*-Value Compared
Subtype A				
Age at resistance test			0.226	
<18 (n = 1)	1 (100%)	0 (0%)		-
19–30 (n = 12)	3 (25.0%)	9 (75.0%)		0.014
31–55 (n = 19)	9 (47.4%)	10 (52.6%)		0.749
>56 (n = 6)	4 (66.7%)	2 (33.3%)		0.247
Recentness of infection			0.203	
Chronic (n = 18)	10 (55.6%)	8 (44.4%)		0.501
Recent (n = 20)	7 (35.6%)	13 (64.4%)		0.011
Viral load at diagnosis (log10 copies/mL)			0.025	
≤4.0 (n = 7)	0 (0%)	7 (100%)		-
4.1–5.0 (n = 13)	7 (53.8%)	6 (46.2%)		0.698
≥5.1 (n = 17)	10 (58.8%)	7 (41.2%)		0.305
Subtype B				
Sex			0.109	
Male (n = 913)	451 (49.4%)	462 (50.6%)		0.608
Female (n = 104)	60 (57.7%)	44 (42.3%)		0.026
Age at resistance test			<0.001	
<18 (n = 4)	0 (0%)	4 (100%)		-
19–30 (n = 245)	89 (36.3%)	156 (63.7%)		<0.001
31–55 (n = 685)	363 (53.0%)	322 (47.0%)		0.026
>56 (n = 88)	61 (69.3%)	27 (30.7%)		<0.001
Region of origin			0.109	
Western Europe (n = 770)	387 (50.3%)	383 (49.7%)		0.200
Eastern Europe (n = 31)	17 (54.8%)	14 (45.2%)		0.450
Africa (n = 10)	7 (70.0%)	3 (30.0%)		0.074
South America (n = 25)	17 (68.0%)	8 (32.0%)		0.011
Other (n = 15)	11 (73.3%)	4 (26.7%)		0.011
Recentness of infection			<0.001	
Chronic (n = 454)	310 (68.3%)	144 (31.7%)		<0.001
Recent (n = 578)	206 (35.6%)	372 (64.4%)		<0.001
Viral load at diagnosis (log10 copies/mL)			<0.001	
≤4.0 (n = 228)	69 (30.3%)	159 (69.7%)		<0.001
4.1–5.0 (n = 390)	167 (42.8%)	223 (57.2%)		<0.001
≥5.1 (n = 397)	270 (68.0%)	127 (32.0%)		<0.001
Migrant status			0.025	
Yes (n = 107)	66 (61.7%)	41 (38.3%)		0.001
No (n = 744)	373 (50.1%)	371 (49.9%)		0.939
Subtype G				
Sex			0.046	
Male (n = 8)	2 (25.0%)	6 (75.0%)		0.046
Female (n = 8)	6 (75.0%)	2 (25.0%)		0.046
Region of origin			0.054	
Western Europe (n = 6)	2 (33.3%)	4 (66.7%)		0.247
Eastern Europe (n = 1)	0 (0%)	1 (100%)		-
Africa (n = 5)	5 (100%)	0 (0%)		-
South America (n = 25)	0 (0%)	1 (100%)		-

## Data Availability

Restrictions apply to the availability of these data. Data were obtained from the EuResist Network and is available on request via a study application form at https://www.euresist.org/become-a-partner (accessed on 16 December 2021) with the permission of the EuResist Network.

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
