# Peer review of "The Role of Late Presenters in HIV-1 Transmission Clusters in Europe"

_viruses, 2023, doi:10.3390/v15122418_

Round 1

Reviewer 1 Report (Previous Reviewer 3)

Comments and Suggestions for Authors

The aim of this study was to evaluate the role of late presenters on transmission.  Some key details need to be clarified.

Methods

-       Need to clarify how the 2679 individuals were chosen for analysis. Random?  Or was it a biased sample. If not a random sample, than it should be made clear that a biased sample could be a limitation of the study

Results -issues needing clafification

-       In Table 2, unclear how a late presented can have recent infection and vice versa

-       The p values would be better if run on Chi-Squared or Fisher exact tests, rather than just comparing between LP and NLP … which Im not even sure how that was done

Discussion –

-       The Discussion is disjointed – it should discusss the results rather than rehashing the results. What do the finding mean…

-       The authors discuss viral load as being higher in clustering LP – authors should make clear when this viral load was collected… at the time of diagnosis?  If so, unclear how NLP would have a undetectable VL.  My guess that VL is a problematic variable in this type of study, as some will be sampled pre-treatment, and others after

-       There is a discussion regarding migration and Brazil – but this was not presented in the results,

-       There are numerous errors in the Discussion that need revision

o   Please have a native English Speaker read and revise the Discussion in particular

o   Line 285. Subtypes listed are A, B, C – but A,B,and G were analyzed in the study

Comments on the Quality of English Language

The Discussion in particular could be improved with some English editing

Author Response

The aim of this study was to evaluate the role of late presenters on transmission.  Some key details need to be clarified.

Methods

-       Need to clarify how the 2679 individuals were chosen for analysis. Random?  Or was it a biased sample. If not a random sample, than it should be made clear that a biased sample could be a limitation of the study

R: We thank the reviewer for the comment. We selected these 2679 from the EuResist database based on the inclusion criteria- diagnosed between 2008-2019, and as stated in lines 90-92: “Only ART-naïve individuals, those who had a sample collection date for first drug resistance test before the date of start of first therapy, were included. “Indeed, it is not a random sample. It is a convenience sampling approach. We have added the clarification in the methods (line: 76) and a note to this in the limitations section (lines 352-354): ‘Despite the large size and coverage of the EuResist database, a major limitation is that it is based on a convenience sampling approach. As such, we must acknowledge potential bias in our results due to the sample used.’

Results -issues needing clafification

-       In Table 2, unclear how a late presented can have recent infection and vice versa

R: The variable recentness of infection was based on the ambiguity rate of the genomic sequences, as stated in lines: 131-133. The definition of late presentation based on CD4 cell count has some flaws known, especially for individuals with an acute infection and major initial drops in CD4 counts which are erroneously classified as LP. We used the ambiguity rate of genomic sequences as a method to test the bias of this definition, since the ambiguity rate of sequences is reportedly inversely correlated to the recentness of infection. In this case, we demonstrate that there is a consistent classification between LP definition and ambiguity rate definition, with a higher proportion of LP being classified as having a “chronic infection” rather than a recent, and vice-versa for NLP, which is accurate with the CD4 cell count definition.

We added a sentence to the discussion (lines:288-292): “One important finding of our study showed incongruences between the definition of LP based on CD4 counts and sequence ambiguity levels that correlate to recentness of infection. While both definitions should reflect recentness of infection, we found some cases of incongruence between the two classifications. These findings should be further considered in later investigations.”

-       The p values would be better if run on Chi-Squared or Fisher exact tests, rather than just comparing between LP and NLP … which Im not even sure how that was done

R: We thank the reviewer for the comment. The p-values for comparing variables (eg. Age vs LP/NLP) were run on Chi-Square or Fisher exact tests, the p-values compared for comparing categories from each variable (eg. chronic infection between LP vs NLP) were run on the epitools website through the “2-sample z-test to compare sample proportion”.

Discussion –

-       The Discussion is disjointed – it should discusss the results rather than rehashing the results. What do the finding mean…

R: We thank the reviewer for the comment. We have re-worked on the discussion, and we consider it much better now.

-       The authors discuss viral load as being higher in clustering LP – authors should make clear when this viral load was collected… at the time of diagnosis?  If so, unclear how NLP would have a undetectable VL.  My guess that VL is a problematic variable in this type of study, as some will be sampled pre-treatment, and others after

R: We thank the reviewer for the comment. The time of the Viral Load is at diagnosis, as stated in the supplementary table 1 and in line 259. Although there was missing the information in some parts of the text, therefore we added the information in table 2 as well as in those parts missing in the text.

-       There is a discussion regarding migration and Brazil – but this was not presented in the results,

R: We thank the reviewer for the comment. In our analysis, we classified the region of origin according to continents (Western Europe, Eastern Europe, South America, Sub-Saharan Africa). We also classified migrant vs native according to their country of origin vs country of follow-up. Herein, we found that migrants from South America were frequently represented among late presenters inside clusters. For that reason, we went to the original dataset to observe which country of origin was more prevalent among those migrants from South America. We found that the higher proportion were migrants originated from Brazil, therefore inside the “South America” region of origin Brazil was the most represented in the sample, and with various reports in the literature about the high concentration of a HIV-1 MSM epidemic, we concluded that our results (more migrants from South America with late presentation found inside clusters), were in accordance with other studies.

-       There are numerous errors in the Discussion that need revision

o   Please have a native English Speaker read and revise the Discussion in particular

R: We have re-worked on the discussion. Thank you for the comment.

  • Line 285. Subtypes listed are A, B, C – but A,B,and G were analyzed in the study

R: We thank the reviewer for the correction. It is already changed in the text.

The Discussion in particular could be improved with some English editing

Reviewer 2 Report (Previous Reviewer 4)

Comments and Suggestions for Authors

I think that the authors addressed all comments from reviewers.

Author Response

I think that the authors addressed all comments from reviewers.

R: We thank the reviewer for revising the manuscript.

Reviewer 3 Report (Previous Reviewer 5)

Comments and Suggestions for Authors

The present study is based on phylogenetic analysis and there are some flaws, most of them are reported to the discussion of your article.

Regardless of the results of the study, the late presenters constitute a potential source of HIV transmission and a public health great concern.

Author Response

The present study is based on phylogenetic analysis and there are some flaws, most of them are reported to the discussion of your article.

Regardless of the results of the study, the late presenters constitute a potential source of HIV transmission and a public health great concern.

R: We thank the reviewer for the comments. We think it reflects very well the main message of our study.

Round 2

Reviewer 1 Report (Previous Reviewer 3)

Comments and Suggestions for Authors

Thank you for your corrections.  I have a few more suggestions

1) Make clear throughout the paper that what you are looking at is not risk of transmission, but risk of phylogenetic clustering (those are different things... as the phylogeny may not reflect transmission)

2) There are still many English edits required, including just in the abstract 3 hyphens and misspellings

3) Please make clear in the abstract as well that this is a convenience sample

4) Sentence structure and flow throughout the manuscript needs significant editing, please send to a English language editor to review

5) There are mentions about sequence ambiguity (likely remaining from the orginal version) but those seem out of place as this is not described at all in this paper

Comments on the Quality of English Language

See above

Author Response

Thank you for your corrections.  I have a few more suggestions

  • Make clear throughout the paper that what you are looking at is not risk of transmission, but risk of phylogenetic clustering (those are different things... as the phylogeny may not reflect transmission)

R: We thank the reviewer for the comment. We have revised the paper and we think that it is now clear. However, we must emphasize that we are looking at HIV-1 transmission clusters, as defined and analysed in previous manuscripts. It a well established methodology in HIV-1 research and, although it does not reflect necessarily direct transmission, it does reflect transmission patterns that are happening in the epidemic.

2) There are still many English edits required, including just in the abstract 3 hyphens and misspellings

R: We thank the reviewer for the comment. We have revised the whole manuscript and think that it is now very much improved.

3) Please make clear in the abstract as well that this is a convenience sample

R: We thank the reviewer for the suggestion. It’s already added to the abstract.

4) Sentence structure and flow throughout the manuscript needs significant editing, please send to a English language editor to review

R: We thank the reviewer for the comment. We have revised the whole manuscript and think that it is now very much improved.

5) There are mentions about sequence ambiguity (likely remaining from the orginal version) but those seem out of place as this is not described at all in this paper

R: We thank the reviewer for the comment. Those pieces of text have been corrected to “recentness of infection (based on sequence ambiguity rate)”

This manuscript is a resubmission of an earlier submission. The following is a list of the peer review reports and author responses from that submission.

Round 1

Reviewer 1 Report

Comments and Suggestions for Authors

General comments:

- This is clearly a large amount of work, with a big dataset obtained from various European countries. It is well written and easy to read.

- As researchers I believe we should give a good example and stop using stigmatizing language such as HIV-infected individuals/patients, rather use individuals living with HIV. Please update this throughout the manuscript, and advocate the use of appropriate language to your scientific colleagues. For guidelines on appropriate terminology there are several good sources available such as https://www.unaids.org/sites/default/files/media_asset/2015_terminology_guidelines_en.pdf

https://www.hptn.org/sites/default/files/inline-files/NIAID%20HIV%20Language%20Guide%20-%20March%202020.pdf 

Specific comments: 

- Introduction: Many factors may influence transmission, and for example the contribution of acute infections is likely through very high viral loads at this stage for which there is some data to support. There was minimal referencing here (only 3 papers, of which one self-citation). Such a complex issue of transmission could be a bit more elaborated in the intro. 

- Methods: it seems that the sequence ambiguity to determine recentness of infection is applied to all sequences, also when the sequences were obtained during ART. This method is already not flawless to determine recency in naive sequences. In the reference mentioned by the authors the sensitivity is 88.8%, specificity only 74.6%, and in this reference paper it's applied only to ART naive sequences. This limitation of only applying to naive sequences is also mentioned by other studies who evaluated this method, such as Meixenberger et al (https://www.ncbi.nlm.nih.gov/pmc/articles/PMC4136167/)   

 In this paper it seems to been applied to therapy experienced individuals, but ART impacts viral diversity. I am not aware of studies that have evaluated this method in sequences taking during therapy failure. Furthermore, if the sequence is obtained during therapy failure (because in order for it to be obtained during therapy the viral load had to unsuppressed) the sequence is already very likely not obtained during the recent infection, but it does not mean the individual was in chronic infection at diagnosis and start of therapy (and thus it does not say anything about for how long this individual has been contributing to transmission).

If there is no reference to a proper evaluation of this method in therapy experienced sequences, I would recommend to only apply this method to treatment naive sequences, and relate this to other available data that could inform recency (date of diagnosis, date of last negative test, CD4 count etc). There is for instance no mentioning of how much time there was between date of diagnosis and timing of sequencing. It is also not clear how the determination of recency based on sequence ambiguity related to determination of late presentation based on CD4 count. 

- Discussion/ conclusion: The aim is to determine factors associated with clustering, and specifically understanding the role of late presentation on transmission. To determine clusters the genetic distance and branch support was used. Late presenters will have longer branches (longer genetic distance) and therefore are already less likely to be part of transmission clusters considering its definition. It has been shown in many papers previously that individuals who were recently infected are more likely to be part of transmission clusters. Also MSM have been described before to be more often part of transmission clusters. In Europe, MSM are among those with a lower risk of being a late presenter. To conclude from the fact that LPs are less often part of a transmission cluster, that late presenters play a limited role on HIV transmission is in my opinion an incorrect interpretation of the data. The method used makes them by definition more likely to be outside of a cluster. There is no mention of this in the discussion. 
And I am not sure how this message is relevant for public health. What does it mean if late presenters don't contribute much to transmission? It is not presented in a way that public health specialists and policy makers could use. The studies on treatment as prevention have so clearly demonstrate the benefit of early diagnosis and early start of therapy, making late presentation such a clear barrier to efficacy of TasP that needs to be addressed. I don't see how this paper would change the view on the importance of late presentation as a barrier towards 95-95-95 and ending the epidemic. 

I think the data on the clusters in Europe for the different subtypes are very interesting and well presented, and perhaps the paper should be more focused on this, rather than the role of late presentation. 

Author Response

General comments:

- This is clearly a large amount of work, with a big dataset obtained from various European countries. It is well written and easy to read.

- As researchers I believe we should give a good example and stop using stigmatizing language such as HIV-infected individuals/patients, rather use individuals living with HIV. Please update this throughout the manuscript, and advocate the use of appropriate language to your scientific colleagues. For guidelines on appropriate terminology there are several good sources available such as https://www.unaids.org/sites/default/files/media_asset/2015_terminology_guidelines_en.pdf

https://www.hptn.org/sites/default/files/inline-files/NIAID%20HIV%20Language%20Guide%20-%20March%202020.pdf 

R: Thank you for your suggestion. The terminology is already corrected.

Specific comments: 

- Introduction: Many factors may influence transmission, and for example the contribution of acute infections is likely through very high viral loads at this stage for which there is some data to support. There was minimal referencing here (only 3 papers, of which one self-citation). Such a complex issue of transmission could be a bit more elaborated in the intro. 

R: We thank the reviewer for the comment. We included a sentence referring to individuals in the acute infection and the likelihood of HIV transmission due to higher viral load values. Lines 55-57 “Furthermore, higher values of viral load on an individual living with HIV can also increase the likelihood of transmission, especially in individuals in the acute phase of infection.”

- Methods: it seems that the sequence ambiguity to determine recentness of infection is applied to all sequences, also when the sequences were obtained during ART. This method is already not flawless to determine recency in naive sequences. In the reference mentioned by the authors the sensitivity is 88.8%, specificity only 74.6%, and in this reference paper it's applied only to ART naive sequences. This limitation of only applying to naive sequences is also mentioned by other studies who evaluated this method, such as Meixenberger et al (https://www.ncbi.nlm.nih.gov/pmc/articles/PMC4136167/)

In this paper it seems to been applied to therapy experienced individuals, but ART impacts viral diversity. I am not aware of studies that have evaluated this method in sequences taking during therapy failure. Furthermore, if the sequence is obtained during therapy failure (because in order for it to be obtained during therapy the viral load had to unsuppressed) the sequence is already very likely not obtained during the recent infection, but it does not mean the individual was in chronic infection at diagnosis and start of therapy (and thus it does not say anything about for how long this individual has been contributing to transmission).

 If there is no reference to a proper evaluation of this method in therapy experienced sequences, I would recommend to only apply this method to treatment naive sequences, and relate this to other available data that could inform recency (date of diagnosis, date of last negative test, CD4 count etc). There is for instance no mentioning of how much time there was between date of diagnosis and timing of sequencing. It is also not clear how the determination of recency based on sequence ambiguity related to determination of late presentation based on CD4 count. 

R: We thank the reviewer you the comment. In our paper we used the sequence ambiguity rate to perform a quality control of the sequences first for ART-naïve and ART-experienced individuals. However, we understand your point-of-view and decided to correct the analysis applying the “recentness of infection” variable only to ART-naïve individual’s. The analysis was corrected in Tables 1 and 2 and Supplementary Tables 1 and 2.. However, it did not change our conclusions, since our findings were highly similar.

In this database we do not have information regarding last negative test. In a previous study of our we correlated sequence ambiguity rate with late presentation to support the late presentation definition based on CD4 cell count. We found that it was inversely correlated, meaning for higher CD4 cell counts, lower the sequence ambiguity rate. If the ambiguity rate is lower, the individual is considered to have a recent infection, and with a higher CD4 cell count it is classified as non-late presenter. These findings are in accordance with each other.

- Discussion/ conclusion: The aim is to determine factors associated with clustering, and specifically understanding the role of late presentation on transmission. To determine clusters the genetic distance and branch support was used. Late presenters will have longer branches (longer genetic distance) and therefore are already less likely to be part of transmission clusters considering its definition. It has been shown in many papers previously that individuals who were recently infected are more likely to be part of transmission clusters. Also MSM have been described before to be more often part of transmission clusters. In Europe, MSM are among those with a lower risk of being a late presenter. To conclude from the fact that LPs are less often part of a transmission cluster, that late presenters play a limited role on HIV transmission is in my opinion an incorrect interpretation of the data. The method used makes them by definition more likely to be outside of a cluster. There is no mention of this in the discussion. 
And I am not sure how this message is relevant for public health. What does it mean if late presenters don't contribute much to transmission? It is not presented in a way that public health specialists and policy makers could use. The studies on treatment as prevention have so clearly demonstrate the benefit of early diagnosis and early start of therapy, making late presentation such a clear barrier to efficacy of TasP that needs to be addressed. I don't see how this paper would change the view on the importance of late presentation as a barrier towards 95-95-95 and ending the epidemic. 

R: We thank the reviewer for the comment. In fact, we have addressed this problem with the methodology by being flexible with the genetic distance threshold used and considering larger genetic distances in the clusters definition. In our analysis, we find that no matter what threshold is used for genetic distance, the statistical analysis always indicates that late presenters are more frequently outside clusters. Therefore, we don’t believe that longer branches in LPs would affect our conclusions.

As the reviewer also said, late presentation is a barrier for the efficacy of TasP, and since there weren’t many studies regarding transmission clusters and late presentation, we wanted to investigate their impact on transmission. Indeed, our findings don’t add much to the importance of late presentation as a barrier towards 95-95-95. It just reinforces the idea that early diagnosis is important given than infection is transmitted early. On the other hand, it allows to understand HIV-1 transmission dynamics in more detail. For example, we found that those late presenters that are inside transmission clusters had higher values of viral load, which could indicate a higher transmissibility of the virus for those individuals inside those clusters.

I think the data on the clusters in Europe for the different subtypes are very interesting and well presented, and perhaps the paper should be more focused on this, rather than the role of late presentation. 

R: Our paper was designed to consider each subtype epidemic separately, as they correspond to separate epidemics. Independently of the subtype epidemic, our findings are quite consistent in terms of clinical determinants of transmission. On the other hand, the demographic determinants of transmission vary between different subtypes. This is what we’ve tried to address and emphasize in this manuscript. We’ll make another analysis later on to investigate more deeply transmission dynamics of the different HIV-1 subtypes.

Reviewer 2 Report

Comments and Suggestions for Authors

Miranda et al evaluated the role of Later presenter (LP) on HIV-1 transmission in Europe. The study includes a huge number of HIV-1 sequences obtained for clinical routine in Europe since 80ths and  it provides important information about the spreading of HIV infection. However, a considerable amount of missing data should be justified and more explanation about the timing of genotyping should be provided. The manuscripts needs several clarifications in methods and results to improve the clarity.

In particular:

Material and methods

1)     Line 86. How did you manage the different length of sequences in phylogenetic analyses? (See also point 3). I suppose that integrase sequence were not considered due to the fact that amplicons obtained for sequencing are not contiguous with PR/RT regions and they do not contribute in improving phylogenetic signals; this should be specified as limitation.

2)     Lines 86-89. This sentence should be clarified. It is not clear if sequences included in the study were from cART naïve and/or cART experienced patients. At this point, to improve clarity, a more clear description of individuals included and the number of sequence per person analysed are needed.  Moreover, a more detailed description of resistance evaluation should be implemented. In fact, it is only indicated the evaluation of transmitted drug resistance at this point but not of acquired drug resistance (ADR).  ADR is reported in results, but without any specification about list of resistance mutations used.

3)     Line 108. In this sentence the length of sequences in final dataset is indicated, but this is not clear after reading sentence at line 86 (see point 1). Please clarify and integrate the information according to previous sentence.

4)     Lines 125-131. Here the authors specify clearly the treatment status, but probably this explanation should be moved or partially repeated above according the comments at point 2.

5)     Line 135-136. Is this definition referred to cART naïve? Did you considered cART experienced individual as LP as default? In supplementary material is indicated that CD cell count is available for 24321 individuals (12501+11820), how did you manage missing values? Considering that the identification of LP and NLP is  the main focus of the study these concerns are crucial. Please clarify.

Results

6)     Lines 150-169. In the description of patients characteristics the information about the year of genotyping is lacking. The time frame indicated in line 75 “38531 HIV-1 infected patients from the EuResist Integrated Database (EIDB) between 1981 and 2019 were included in this study” is very large and generic. Dynamics of HIV transmission might change over time, how the authors managed this issue? Sensitivity analyses on the subgroup of individuals genotyped for first time in the last 5 year might be important to check if the global results are consistent to those we are currently observing. Please clarify.

7)     Lines 257-258. “The proportion of late presenters and non-late presenters in clusters was analyzed  and LP were more in small clusters (95·7%) than NLP (92·4%)”, the values 95.7 vs. 92.2 are similar, please check the sentence.

8)     Lines 255-265. In paragraph 3.6 “Transmission Clusters analysis:” I suppose that denominator of proportions is 8335, but it should be indicated also in figure 4 A. The third column of Figure 4A indicates “<8 patients”. Do these numbers included also cluster with 2 patients? Or Should it be indicated as “3-8 patients? Please clarify.

9)     Figure 4B. Please indicate the denominators for subtype A, B and G in the figure.

10)  Table 2. The numbers in these table are unclear, because of missing data which are not indicated. In fact, the sum of patients per each variable is different, e.g. for subtype A (total number missing) the total number of patients according to sex is 173 (109+64), to Age 111 (4+33+62+56), to recentness of infection 179 (84+95). The amount of missing data indicated in table S1 is considerable, how the authors managed this issue? Authors should argue about this concern as limitation of the work and/or include missing values per each variable as “unknown” in table 2. The denominators for LP and NLP should be included to improve the clarity (see also point 5). Please carefully revise.

Discussion

11)  As indicated above the concerns about the considerable amount of missing data should be better discussed as study limitation.

Author Response

Miranda et al evaluated the role of Later presenter (LP) on HIV-1 transmission in Europe. The study includes a huge number of HIV-1 sequences obtained for clinical routine in Europe since 80ths and  it provides important information about the spreading of HIV infection. However, a considerable amount of missing data should be justified and more explanation about the timing of genotyping should be provided. The manuscripts needs several clarifications in methods and results to improve the clarity.

In particular:

Material and methods

  • Line 86. How did you manage the different length of sequences in phylogenetic analyses? (See also point 3). I suppose that integrase sequence were not considered due to the fact that amplicons obtained for sequencing are not contiguous with PR/RT regions and they do not contribute in improving phylogenetic signals; this should be specified as limitation.

R: We thank the reviewer for the comment. Indeed, we did not use integrase sequence, we just used the regions of RT and PR. We added a sentence (lines: 388- 390) “Another limitation of our study was the exclusion of the integrase sequences from the analyses, since this region of the sequences does not contribute to the improvement of phylogenetic signals ” in the section on limitations of the study as well.

  • Lines 86-89. This sentence should be clarified. It is not clear if sequences included in the study were from cART naïve and/or cART experienced patients. At this point, to improve clarity, a more clear description of individuals included and the number of sequence per person analysed are needed.  Moreover, a more detailed description of resistance evaluation should be implemented. In fact, it is only indicated the evaluation of transmitted drug resistance at this point but not of acquired drug resistance (ADR).  ADR is reported in results, but without any specification about list of resistance mutations used.

R: We thank the reviewer for the comment. The classification of ART-naïve and ART-experienced individuals is already added in lines 88-91. In the lines 92-96 we specify that only the first genomic sequence per patient was used. For ADR, the definition on how we classified and how we calculated is now added in lines 100-101.

  • Line 108. In this sentence the length of sequences in final dataset is indicated, but this is not clear after reading sentence at line 86 (see point 1). Please clarify and integrate the information according to previous sentence.

R: We thank the reviewer for the comment. In lines 88-89 we indicate that the inclusion criteria was for a length of sequences between 500-1000 nucleotides. In lines 118-119 we indicate the mean length of our dataset.

  • Lines 125-131. Here the authors specify clearly the treatment status, but probably this explanation should be moved or partially repeated above according the comments at point 2.

R: We added an explanation in lines 88-91 similar to the one in the study variables to characterize the classification of ART-naïve and ART-experienced individuals.

  • Line 135-136. Is this definition referred to cART naïve? Did you considered cART experienced individual as LP as default? In supplementary material is indicated that CD cell count is available for 24321 individuals (12501+11820), how did you manage missing values? Considering that the identification of LP and NLP is  the main focus of the study these concerns are crucial. Please clarify.

R: We thank the reviewer for the comment. The definition of LP was based on the CD4 cell count at diagnosis of individuals living with HIV. We considered both ART-naïve and ART-experienced individuals. Table 2 shows the characteristics of LP and NLP in clusters. For this analysis we only included individuals in cluster and with information regarding their late presentation status. Regarding the table S1, we considered for each variable the absolute number of individuals with available information. For that reason, we stated that the total number of included individuals was 38531, which is equal to 100%. Then, for each variable, the absolute number of individuals with the available information was calculated based on the difference between the total number (38531) and the number available (eg: CD4 cell count= 24321). Then, for the specific variable, we considered that that number was the variable 100%, this means that the information from 24321 individuals, which according to all included individuals accounted for 63.1%, but for calculations for the variable itself accounted for 100%.

Results

  • Lines 150-169. In the description of patients characteristics the information about the year of genotyping is lacking. The time frame indicated in line 75 “38531 HIV-1 infected patients from the EuResist Integrated Database (EIDB) between 1981 and 2019 were included in this study” is very large and generic. Dynamics of HIV transmission might change over time, how the authors managed this issue? Sensitivity analyses on the subgroup of individuals genotyped for first time in the last 5 year might be important to check if the global results are consistent to those we are currently observing. Please clarify.

R: We thank the reviewer for the comment. The reviewer is correct. In another future analysis, we will look at the clusters of transmission across time. However, here it was not possible to use a molecular clock given the large amount of data considered.

  • Lines 257-258. “The proportion of late presenters and non-late presenters in clusters was analyzed  and LP were more in small clusters (95·7%) than NLP (92·4%)”, the values 95.7 vs. 92.2 are similar, please check the sentence.

R: We thank the reviewer for the comment. We have changed the sentence to (lines:264-265):

The proportion of late presenters and non-late presenters in clusters was analyzed and LP were slightly more in small clusters (95·7%) than NLP (92·4%).”

8)     Lines 255-265. In paragraph 3.6 “Transmission Clusters analysis:” I suppose that denominator of proportions is 8335, but it should be indicated also in figure 4 A. The third column of Figure 4A indicates “<8 patients”. Do these numbers included also cluster with 2 patients? Or Should it be indicated as “3-8 patients? Please clarify.

R: We thank the reviewer for the comment. It is already clarified in the figure and in the legend of the figure.

  • Figure 4B. Please indicate the denominators for subtype A, B and G in the figure.

R: We thank the reviewer for the comment. The denominators are already included in the figure 4B.

  • Table 2. The numbers in these table are unclear, because of missing data which are not indicated. In fact, the sum of patients per each variable is different, e.g. for subtype A (total number missing) the total number of patients according to sex is 173 (109+64), to Age 111 (4+33+62+56), to recentness of infection 179 (84+95). The amount of missing data indicated in table S1 is considerable, how the authors managed this issue? Authors should argue about this concern as limitation of the work and/or include missing values per each variable as “unknown” in table 2. The denominators for LP and NLP should be included to improve the clarity (see also point 5). Please carefully revise.

R: We thank the reviewer for the comment. As stated above, Table 2 shows the characteristics of LP and NLP in clusters. For this analysis we only included individuals in cluster and with information regarding their late presentation status. Regarding the table S1, we considered for each variable the absolute number of individuals with available information. For that reason, we stated that the total number of included individuals was 38531, which is equal to 100%, for each variable the absolute number of individuals with the available information was calculated based on the difference between the total number (38531) and the number available (eg: sex= 36699). Then, for the specific variable, we considered that that number was the variable 100%, this means for example for sex there was information from 36699 individuals, which according to all included individuals accounted for 95.2%, but for calculations for the variable itself accounted for 100%. The denominators for LP and NLP for each subtype were added to Table 2.

Also, the missing data limitation was added to the limitations section lines: 384-388

Discussion

  • As indicated above the concerns about the considerable amount of missing data should be better discussed as study limitation.

R: We thank the reviewer for the comment. A sentence regarding this topic was also added in the limitation section (lines 384-388:) “The use of the EuResist database has an important limitation concerning the considerable amount of missing data for some variables. However, given the very large size of the dataset and integration of different types of data, among the largest available worldwide, the statistical power in the analysis of the dataset is still very high”.

Reviewer 3 Report

Comments and Suggestions for Authors

The authors present a large phylogenetic analysis of several national European epidemics to get at the question of what is the contribution of late presenters to ongoing transmission.  While this is a hugely important topic, I dont think the analysis the authors present here really gets at that question, and most importantly when presenting their results, they do not discuss many of the key assumptions and pitfalls associated with the analysis

- The definition of late presenter chosen (CD4<350) while from a consensus definition, is known to be imperfect (persons with acute infection can also present with CD4 counts <350) and this should be presented in the discussion

- The use of ambiguity to assess recency also has issues, and it would be important to note that all of the cohorts are using the same approach to defining ambiguities

- Clustering is not a proxy for transmission as the authors are alluding. It can be also a function of sampling… for which there is sure to be a difference across the analyzed time period from 1981-2019. Between 1981 -2019, sampling depth and timing has likely varied signficantly over this period, and that will impact clustering and identification of late presenter.  Consider a narrower window of analysis when sampling may be more consistent and uniform Could consider either splitting the analysis into time periods, or focusing on the most recent 5y

- This is highlighted by the finding that PWIDs are less likely to cluster. It is well known that IDU outbreaks and transmissions have more similar sequences given reduced evolution during a IDU transmission

- The finding that LP cluster less may be just because they are sampled further from the time of infection, and may have nothing to do with transmission

- These  pitfalls and assumptions should be described in the discussion 

- The discussion spends a great deal of space rehashing results… can focus on what new info this study brings, and the limitations

-As per my comments above… conclusions need to be appropriately toned down

Minor points

-line 108  - is this median or mean length

Comments on the Quality of English Language

English should be reviewed by a native English speaker, there is phrasing that should be simplifeid

Author Response

The authors present a large phylogenetic analysis of several national European epidemics to get at the question of what is the contribution of late presenters to ongoing transmission.  While this is a hugely important topic, I don’t think the analysis the authors present here really gets at that question, and most importantly when presenting their results, they do not discuss many of the key assumptions and pitfalls associated with the analysis

- The definition of late presenter chosen (CD4<350) while from a consensus definition, is known to be imperfect (persons with acute infection can also present with CD4 counts <350) and this should be presented in the discussion

R: We thank the reviewer for the comment. A sentence was added in the limitations section (lines:374-379) with reference to that topic lines: “Our work focused mainly on the comparison between LP vs NLP populations. For that purpose, we used the current consensus definition of LP. However, we acknowledge the controversy around this definition, since it is known that when an individual is acutely infected, the CD4 cell count drops before becoming higher again, which means that if he is diagnosed in that time period, i.e. acute stage, the patient can be misdiagnosed as late presenter.”

- The use of ambiguity to assess recency also has issues, and it would be important to note that all of the cohorts are using the same approach to defining ambiguities

R: We thank the reviewer for the comment. All cohorts are using the same approach to defining ambiguities, because since we are analyzing the EuResist database as a whole, we defined and performed the analysis of ambiguity rate. So, in our case, the cohorts used for the analysis, all went through the same approach.

- Clustering is not a proxy for transmission as the authors are alluding. It can be also a function of sampling… for which there is sure to be a difference across the analyzed time period from 1981-2019. Between 1981 -2019, sampling depth and timing has likely varied signficantly over this period, and that will impact clustering and identification of late presenter.  Consider a narrower window of analysis when sampling may be more consistent and uniform Could consider either splitting the analysis into time periods, or focusing on the most recent 5y

R:  We thank the reviewer for the comment. The reviewer is correct. In another future analysis, we will look at the clusters of transmission across time. However, here it was not possible to use a molecular clock given the large amount of data considered.

- This is highlighted by the finding that PWIDs are less likely to cluster. It is well known that IDU outbreaks and transmissions have more similar sequences given reduced evolution during a IDU transmission

R: We are not sure about this. In fact, we see higher OR for MSMs being in clusters both in subtypes A and B. This is clear for subtype B, given that transmission of subtype B in IDUs has largely reduced along time. For subtype, on the other hand, there is frequently transmission of subtype A between MSM both in Western and Eastern Europe, while transmission of subtype A among IDU is mostly limited to Eastern Europe/Russia. Globally, when we crudely look at transmission of subtype A, this could contribute to the higher OR for MSM. However, we added to the limitation section a sentence considering the fact that sampling bias is a limitation for the definition of clusters, lines 382-384: “Furthermore, the accuracy of reconstructed transmission clusters is obviously affected by sampling issues and by the methods used for reconstruction of phylogenetic trees”.

- The finding that LP cluster less may be just because they are sampled further from the time of infection, and may have nothing to do with transmission

R: We thank the reviewer for the comment. In fact, we have addressed this problem with the methodology by being flexible with the genetic distance threshold used and considering larger genetic distances in the clusters definition. In our analysis, we find that no matter what threshold is used for genetic distance, the statistical analysis always indicates that late presenters are more frequently outside clusters. Therefore, we don’t believe that longer branches in LPs would affect our conclusions.

- These  pitfalls and assumptions should be described in the discussion 

R: We thank the reviewer for the comments. The limitation section was updated according to the comments raised by the reviewer, which we agree and added the main concerns and limitations we found during the analysis and writing of the manuscript. (Lines:274-390)

- The discussion spends a great deal of space rehashing results… can focus on what new info this study brings, and the limitations

R: We thank the reviewer for the comment. In our discussion we compare our results with those present in the literature and at the same time we try to give some explanation of our results. Although, we added a sentence highlighting the main findings of this study in the conclusion section (lines:367-370), and we also included more information in the limitations section as we stated in the comment above (lines:274-390)

-As per my comments above… conclusions need to be appropriately toned down

R: We thank the reviewer for the comment. The conclusions were revised as shown in the lines: 367-372.

Minor points

-line 108  - is this median or mean length

R: We thank the reviewer for the comment. It is the mean length. We have clarified it in the text lines 118-119.

Reviewer 4 Report

Comments and Suggestions for Authors

This is a very good paper on TCs and LP in PLWH.

I have 2 major issues:

a) on page 10, table 2: the table is quite dense, I would clearly write out for all subtypes (in the same line of Subtypes A - B - G) the number of LP and NLP in TCs and in brackets the relative % of LP and NLP in TCs vs all LP and all NLP, respectively.

b) on page 12, line 350: add a comment on viral loads of LP outside TCs.

One minor issue: correct the spelling of "Standford" on page 2, line 91.

Author Response

This is a very good paper on TCs and LP in PLWH.

I have 2 major issues:

  1. on page 10, table 2: the table is quite dense, I would clearly write out for all subtypes (in the same line of Subtypes A - B - G) the number of LP and NLP in TCs and in brackets the relative % of LP and NLP in TCs vs all LP and all NLP, respectively.

R: We thank the reviewer for the comment. We added, in table 2, for each subtype the number of LP and NLP inside cluster.

  1. on page 12, line 350: add a comment on viral loads of LP outside TCs.

R: We thank the reviewer for the comment. We studied LP inside clusters only, but according to a previous study where we studied the factors associated with late presentation, we found that usually late presenters have higher values of viral loads when compared to non-late presenters, and we have commented accordingly in lines 351 and 352 .However, to answer your comment we revised the viral loads outside TCs and as our previous study, LP also have higher values of viral load when compared to non-late presenters.

One minor issue: correct the spelling of "Standford" on page 2, line 91.

R: We thank the reviewer for the correction. It is already corrected.

Reviewer 5 Report

Comments and Suggestions for Authors

Comments on the Quality of English Language

Author Response

  1. The Database for so many HIV patients through several decades, since the start of the pandemic, is very impressive and crucial for understanding the origin and evolution of HIV infection in the western Europe.
    However, the attempt to correlate genomic information with clinical, socio- demographic is complicated.

R: We thank the reviewer for the comment. The information present in the database, whether genomic or clinical or socio-demographic, is information gathered through follow-up of patients. The analysis was done by the information collected from this dataset was from sampling times close to each other if not in the same date.

  1. In contrast to the evidence based knowledge that early diagnosis and direct initiation of ART is correlated with a favorable long term outcome for the infected individual and decrease to further transmissions and control to the pandemic, your conclusion is in contrast to this well documented evidence, Showing that late presenters play a small role to HIV-1 transmission.

R: We thank the reviewer for the comment. We agree with the fact that early diagnosis is associated with better clinical outcomes and decrease of HIV-1 transmission, however there is no clear evidence saying that the late presenters have indeed a considerable impact on HIV transmission, there is only evidence indicating that those who present late to diagnosis could have worst clinical outcomes and more difficulties in direct ART initiation and suppressed viral load. In our study, what was observed is that LP are less likely to be inside transmission clusters, what could indicate that they have a limited role in HIV-1 transmission.

  1. The above confusing message is in part due to the definition consensus of late presenters, which is usually used. However, the remaining patients ( with >350 CD4 cell counts) are not fullfil the criterion for early presenters, leading to a disturbing conclusion.

R: We thank the reviewer for the comment. In this database we do not have information regarding last negative test. In a previous study of our we correlated sequence ambiguity rate with late presentation to support the late presentation definition based on CD4 cell count. We found that it was inversely correlated, meaning for higher CD4 cell counts, lower the sequence ambiguity rate. If the ambiguity rate is lower, the individual is considered to have a recent infection, and with a higher CD4 cell count it is classified as non-late presenter. These findings are in accordance with each other.

  1. Since there are many iv drug user in Spain and Portugal, this risk group is under represented to the present study. Moreover, risk groups are often misreported due to stigma and religious, professional and other issues.

R: We thank the reviewer for the comment. I think there is an error, since in reality IVDUs population has largely been reduced in the last 20 years and we have very few IVDUs infected with HIV-1 in this region. The main region where we find a higher proportion of IVDU in our study is in the Eastern European region.

  1. The question about different subtypes and capability of transmittion has not been well answered and defined and probably rather the risk behaviour rather than subtypes determine the infectiousness ( chemsex, iv drug users, multiple sexual partners etc).

R: Indeed, risk behavior is a major contributor to transmission dynamics. However, given that different subtypes correspond to independent epidemics, we have separated the analyses of different subtypes. On the other hand, it is well documented that some subtypes are highly associated to types of transmission and specific vulnerable populations. For example, it is well documented that subtype B is most widely spread, and especially in Brazil among MSM. As for subtype A is highly associated to a IVDU transmission route in eastern European countries.  

  1. Transmission rate can be considered a valuable measure for public health, and provides a proxy of the rate of epidemic growth within a cluster and, therefore, it can be useful for targeted HIV prevention programs.

R: We thank the reviewer for the comment. We agree with the reviewer, when characterizing a transmission cluster, we can highlight some vulnerable populations and the expansion of transmission. In that way, it is easier to target them and develop some population-centered HIV prevention programs.

  1. Since the dataset provides information about transmitted primary resistance before and after 1996 or even later ( with the availability of one pill fixed combination regimens and their impact to halt transmission of HIV and resistance mutations to several class of ART.

R: We thank the reviewer for the comment and we agree with the reviewers comment.

Round 2

Reviewer 1 Report

Comments and Suggestions for Authors

The authors have in my opinion sufficiently addressed the comments

Reviewer 2 Report

Comments and Suggestions for Authors

After the revision of the issues indicated before, I have no further comments.

Reviewer 3 Report

Comments and Suggestions for Authors

I appreciate the author's responses to the my comments, but I do not think they have adequately addressed my major concerns that were also pointed out by other reviewers. The most important of which is the impact of variations in sampling over time. Indeed lower sampling of new diagnoses in the past would automatically make late presenters less likely to cluster.

When responding to the suggestion about using a narrower (and hopefully more consistent) sampling window, the authors have suggested that they could not perform a molecular clock analysis due to the size of the dataset. I was actually not suggesting this.  I was suggesting that instead of looking from 1981 to the present, perhaps only looking at the last 5-10 years, and only naive individuals would provide a more consistent sampling scenario.. and I believe several other reviewers had the same thought